# Ocean acidification increases susceptibility to sub-zero air temperatures in ecosystem engineers and limits poleward range shifts

Jakob Thyrring[1,2]*, Colin D Macleod[1,3], Katie E Marshall[1], Jessica Kennedy[1], Réjean Tremblay[4], Christopher DG Harley[1,5]

[1]Department of Zoology, University of British Columbia, Vancouver, Canada; [2]Department of Ecoscience – Marine Ecology & Arctic Research Centre, Aarhus University, Aarhus C, Denmark; [3]Department of Biological Sciences, University of Alberta, Edmonton, Canada; [4]Institut des sciences de la mer, Université du Québec à Rimouski, Rimouski, Canada; [5]Institute for the Oceans and Fisheries, University of British Columbia, Vancouver, Canada

*For correspondence: thyrring@ecos.au.dk

Competing interest: The authors declare that no competing interests exist.

**Abstract** Ongoing climate change has caused rapidly increasing temperatures and an unprecedented decline in seawater pH, known as ocean acidification. Increasing temperatures are redistributing species toward higher and cooler latitudes that are most affected by ocean acidification. While the persistence of intertidal species in cold environments is related to their capacity to resist sub-zero air temperatures, studies have never considered the interacting impacts of ocean acidification and freeze stress on species survival and distribution. Here, a full-factorial experiment was used to study whether ocean acidification increases mortality in subtidal *Mytilus trossulus* and subtidal *M. galloprovincialis*, and intertidal *M. trossulus* following sub-zero air temperature exposure. We examined physiological processes behind variation in freeze tolerance using [1]H NMR metabolomics, analyses of fatty acids, and amino acid composition. We show that low pH conditions (pH = 7.5) significantly decrease freeze tolerance in both intertidal and subtidal populations of *Mytilus* spp. Under current day pH conditions (pH = 7.9), intertidal *M. trossulus* was more freeze tolerant than subtidal *M. trossulus* and subtidal *M. galloprovincialis*. Conversely, under low pH conditions, subtidal *M. trossulus* was more freeze tolerant than the other mussel categories. Differences in the concentration of various metabolites (cryoprotectants) or in the composition of amino acids and fatty acids could not explain the decrease in survival. These results suggest that ocean acidification can offset the poleward range expansions facilitated by warming and that reduced freeze tolerance could result in a range contraction if temperatures become lethal at the equatorward edge.

## Editor's evaluation

Thyrring et al. provide convincing experimental results on the role of ocean acidification on the survival of two bivalve species. This novel work is fundamental in setting a more mechanistic understanding of the impacts of climate change on ocean species' poleward re-distribution across the globe. The major strength of this work is their usage of state-of-the-art techniques (such as metabolomics, fatty acid and amino acid analysis) to link physiological level processes to global climate change.

## Introduction

The rapid rise in atmospheric $CO_2$ concentration since the industrial revolution has increased global air and water temperatures and caused ocean pH to decline (a process termed ocean acidification) at rates unprecedented in geologic history (*Hönisch et al., 2012*). These environmental changes are causing species range shifts and cascading ecological effects across the globe, resulting in regime shifts and alteration of food web structure (*Kortsch et al., 2012*; *Thyrring et al., 2021*; *Wernberg et al., 2016*). For example, the fish assemblage around the Svalbard archipelago, located in the Arctic Ocean (78°N), is borealizing as Arctic species have retracted northward to cooler areas while boreal species have become dominant (*Fossheim et al., 2015*). Co-occurring ocean acidification is, furthermore, predicted to have severe consequences for marine organisms and communities, and a large body of research has shown a wide range of negative effects. Decreased pH weakens shell production (*MacLeod and Poulin, 2015*) and increases dissolution in calcifying organisms, which are therefore generally more vulnerable to ocean acidification compared to other organisms (*Kroeker et al., 2010*). Ocean acidification has also been found to increase heart rates in some invertebrate species (*Lim and Harley, 2018*) and alter benthic community structure (*Brown et al., 2018*). Elevated temperatures and ocean acidification have furthermore been observed to interact in various ways, causing heterogenic physiological responses across species, depending on taxon and life stage (*Harvey et al., 2013*). Indeed, these two stressors may disproportionally alter species interactions and biodiversity in marine ecosystems (*Franzova et al., 2019*; *Nagelkerken and Munday, 2016*).

While the vast majority of ocean acidification and climate change research has focused on lower latitude systems, studies have rarely considered the impacts on species at their poleward range edge. The poleward edge of subtidal ectotherms (ectotherms that are constantly submerged in water) is determined by low water temperatures (*Sunday et al., 2012*), however, the distribution of intertidal species is also controlled by their capacity to tolerate freezing as they are exposed to sub-zero air temperatures during emersion at high latitudes (*Kennedy et al., 2020*; *Reid and Harley, 2021*; *Thyrring et al., 2019*; *Wang et al., 2020*). On rocky shores, a mosaic of stressors determines biological patterns (*Thyrring and Peck, 2021*). For instance, canopy-forming macroalgae shelter the understory communities from extreme sub-zero air temperatures (*Sejr et al., 2021*), and where cold enough, an ice foot forms on the rocky surface, creating a warmer protective microhabitat increasing survivorship of intertidal organisms residing below (*Scrosati and Eckersley, 2007*; *Thyrring et al., 2017a*). However, as temperatures increase at the northern range edge where ice forms, organisms face sub-zero air temperatures when emerged at low tides as the ice foot melts or before it forms, offsetting the otherwise facilitative effect of ocean warming on range expansions.

Freezing can result in osmotic stress, dehydration, and structural damage to the cell membrane (*Meryman, 1971*; *Storey and Storey, 1988*). While the underlying physiological processes remain poorly understood in intertidal species (*Kennedy et al., 2020*), generally, freeze tolerance mechanisms include accumulation of cryoprotectants (such as amino acids, polyols, or sugars) to protect proteins and membranes (*Denlinger and Lee, Jr, 2010*), and prevent intracellular osmotic stress as water is lost to the extracellular space (*Storey and Storey, 1996*; *Toxopeus and Sinclair, 2018*). For instance, the amino acid proline increases freeze tolerance in plants and insects (*Patton et al., 2007*; *Storey and Storey, 1996*), and the earthworm *Dendrobaena octaedra* survive freezing during winter by accumulating glucose (*Holmstrup, 2003*). In intertidal bivalves, a mixture of metabolites (i.e. low molecular weight compounds), such as trimethylamine n-oxide (TMAO), betaine, strombine, and the amino acid taurine, likely acts as cryoprotectants that increase freeze tolerance (*Kennedy et al., 2020*; *Loomis et al., 1988*). While underexplored, it also appears that many intertidal species may have an array of ice binding proteins that help manage ice growth and propagation (*Box et al., 2022*).

Freeze tolerance in some ectotherms is also associated with the composition of the cell membrane phospholipid fatty acids, which are sensitive to temperature variation (*Hazel, 1995*). Functional membranes must exist in a fluid liquid-crystalline phase maintained by the composition of the phospholipids. Low temperatures decrease membrane fluidity, and the membrane becomes partly dysfunctional, losing selective properties and leaking cell contents (*Hazel, 1995*). Ectotherms can counteract this effect by lipid remodeling and adjustments of cholesterol levels. This mechanism, termed homeoviscous adaptation, has been shown in a wide range of marine and terrestrial animals (*Storey and Storey, 1988*), and intertidal bivalves can remodel phospholipids in response to temperature changes (*Pernet et al., 2007*; *Thyrring et al., 2017c*; *Williams and Somero, 1996*).

Despite this progress on the mechanisms of cold and freeze tolerance in intertidal species, it is completely unknown whether ocean acidification interacts with cryoprotectant production or lipid remodeling. High-latitude cold water is able to absorb significantly more $CO_2$ than lower-latitude warmer water, and therefore seawater pH is decreasing most rapidly at these latitudes (*Fassbender et al., 2017*). Ocean acidification decreases pH in the physiological fluids of osmoconformers with a low capacity to regulate internal pH levels (e.g. bivalves), which could lead to disruption of cellular processes, and shifts in osmotic balance (*Wittmann and Pörtner, 2013*; *Zhao et al., 2020*). Thus, ocean acidification may decrease freeze tolerance and increase animal vulnerability to sub-zero air temperature exposure, yet the interaction between ocean acidification and freeze tolerance interactions remains to be explored.

Bivalves of the genus *Mytilus* are distributed in intertidal habitats in both the Northern and Southern Hemisphere (*Hilbish et al., 2000*; *Mathiesen et al., 2017*; *Thyrring et al., 2017b*). *Mytilus* sp. are commercially and ecologically important ecosystem engineers that create habitats for a diverse associated fauna and are widely used as model organisms for studying impacts of various stressors (*Barrett et al., 2022*; *Telesca et al., 2019*; *Thyrring et al., 2015*). *Mytilus* spp. can survive tissue freezing and are expanding at higher latitudes in response to global warming (*Thyrring et al., 2017a*); however, the performance of *Mytilus* sp. at their poleward edge remains poorly understood. The focus of this study is two *Mytilus* spp. found in British Columbia, Canada; the invasive Mediterranean mussel *Mytilus galloprovincialis* and the native bay mussel *Mytilus trossulus*, allowing a comparison of responses among native and invasive species. By investigating the effects of ocean acidification on freeze tolerance in these species, we test the hypothesis that ocean acidification will generally increase mortality in intertidal species living near their poleward range edge due to an increased susceptibility to sub-zero air temperatures during emersion. Mussels from both the intertidal and subtidal realm were investigated to detect whether previous exposure to air has any effects on freeze tolerance. Specifically, we predict that (1) intertidal animals are more freeze tolerant than subtidal conspecifics, (2) native *M. trossulus* is more freeze tolerant than *M. galloprovincialis,* and (3) ocean acidification will increase freeze mortality in both species. Finally, we hypothesize that variation in freeze tolerance will correlate with (1) a destabilized cell membrane caused by variation in the unsaturation state of

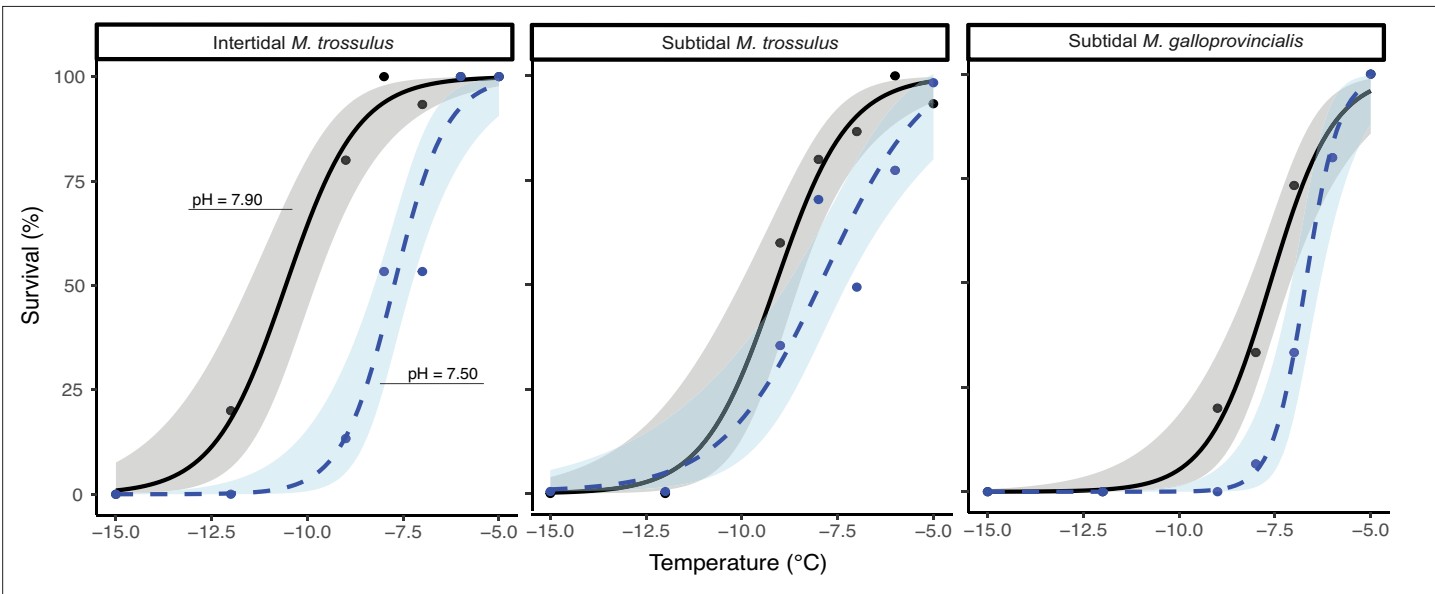

**Figure 1.** Proportion of survival in intertidal *Mytilus trossulus,* subtidal *M. trossulus,* and subtidal *M. galloprovincialis* after 10 days of acclimation to pH conditions (pH = 7.9 or pH = 7.5). Groups of mussels were exposed to seven sub-zero air temperatures. Lines indicate fitted logistic regression models; solid black lines represent control conditions (pH = 7.9); and dashed blue lines represent acidified conditions (pH = 7.5). Dots represent actual survival, and shaded areas indicate 95% confidence intervals of the fitted model.

The online version of this article includes the following figure supplement(s) for figure 1:

**Figure supplement 1.** Mussels placed in wells drilled into a cooled aluminum block to assay freezing survival.

membrane phospholipids, and (2) variation in the composition and concentration of selected molecular cryoprotectants.

# Results

## Survival

Following acclimation to two pH conditions (pH = 7.9 or pH = 7.5), mussels were exposed to seven sub-zero air temperatures (−5,−6, −7,−8, −9, −12, −15°C), and the supercooling point (SCP: indication of internal ice formation) was determined. Internal ice formation was observed in all mussels. There was no significant effect of pH condition (ANOVA; p>0.05) or mussel categories (subtidal *M. trossulus* and *M. galloprovincialis*, and intertidal *M. trossulus*; ANOVA; p>0.05) on the SCP.

We investigated the effect of pH and sub-zero air temperature on survival using generalized linear models (GLMs). There was no significant interaction between the effect of pH and air temperature on any mussel category, and the interaction term was excluded in the final models. Lower sub-zero air temperature significantly decreased the survival of mussels in all three categories exposed to both control and low pH conditions (*Figure 1*; *Supplementary file 1*). Under control conditions (pH = 7.9), the lower lethal temperature at which 50% of the population perished ($LLT_{50}$) was significantly lower in intertidal *M. trossulus* (−10.56°C ± 0.80 CI) compared to subtidal *M. trossulus* (−9.12°C ± 0.48 CI) and subtidal *M. galloprovincialis* (−7.62°C ± 0.49 CI), which was the least freeze-tolerant species (*Figure 1*). Following exposure to low pH (pH = 7.5), survival significantly decreased after sub-zero air exposure in all three mussel categories (*Figure 1*; *Supplementary file 1*). Accordingly, the $LLT_{50}$ of intertidal *M. trossulus* was −7.53°C ± 0.26 CI, while the $LLT_{50}$ was −8.04°C ± 0.32 CI and −6.69°C ± 0.17 CI for subtidal *M. trossulus* and subtidal *M. galloprovincialis*, respectively. Thus, subtidal *M. trossulus* was the most freeze-tolerant category under low pH conditions, although it was not statistically different from intertidal *M. trossulus*. It should be noted that only one *M. galloprovincialis* (6.66%) survived exposure to −8°C.

## Metabolomics and fatty acids

Metabolite, amino acid, and fatty acid analyses were performed on mussels collected after 10 days of pH exposure. We compared the composition of metabolites using principal component analysis (PCA) plots, which showed that the three mussel categories clustered together, suggesting no differentiation in their metabolite profiles under control and acidified conditions (*Figure 2*). The predominate

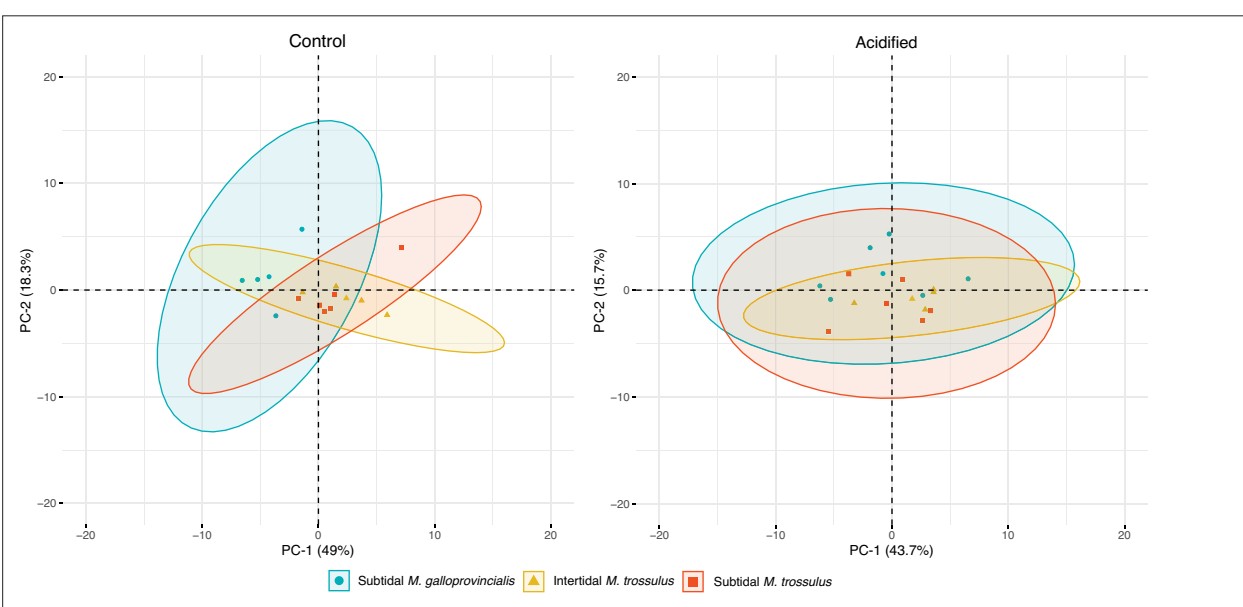

**Figure 2.** Principal component analysis (PCA) plot based on all identified metabolites in intertidal *Mytilus trossulus*, subtidal *M. trossulus,* and subtidal *M. galloprovincialis* after 10 days of acclimation to pH conditions (pH = 7.9or pH = 7.5). Each point represents an individual, and the ellipses extend to the 95% confidence interval of the mussel category.

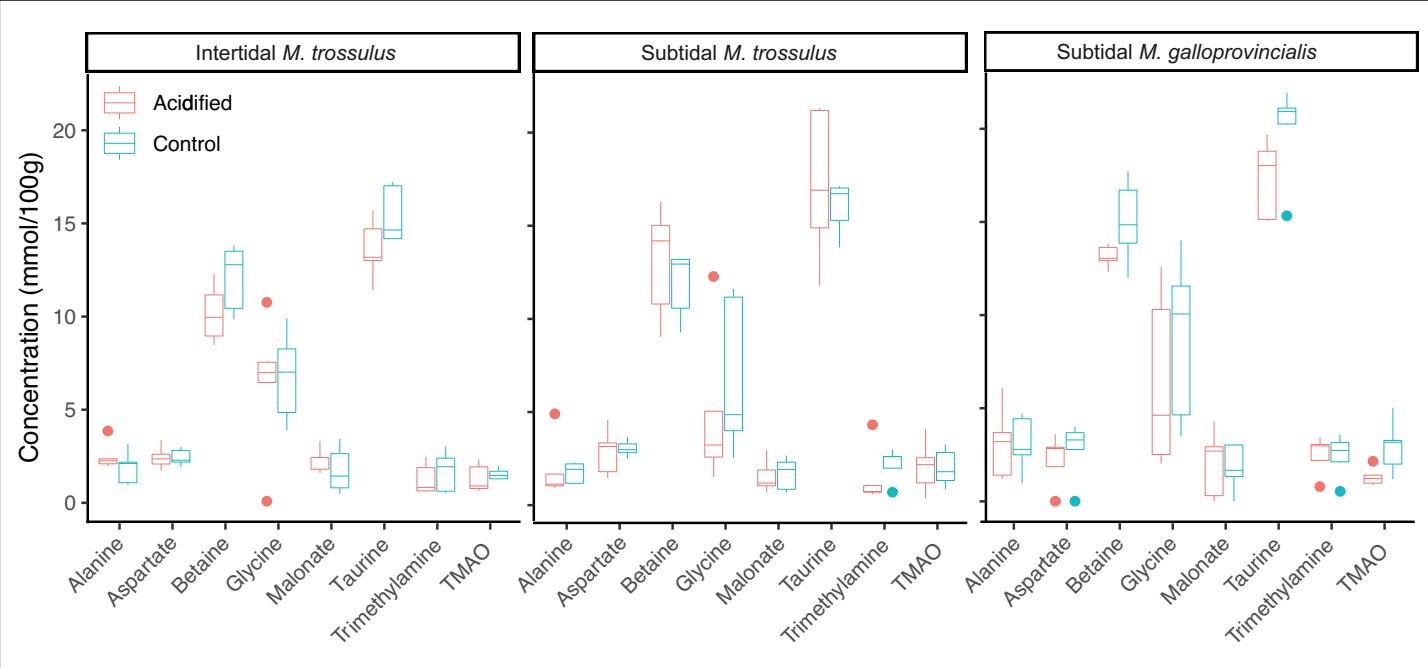

**Figure 3.** The concentration of common gill tissue osmolytes in intertidal *Mytilus trossulus,* subtidal *M. trossulus,* and subtidal *M. galloprovincialis* after 10 days of acclimation to pH conditions (pH = 7.9 or pH = 7.5). The horizontal line in each boxplot is the median, the boxes define the hinges (25–75% quartile), and the whisker is 1.5 times the hinges (n = 5). Colored dots represent data outside this range.

osmolytes were alanine, aspartate, betaine, glycine, malonate, taurine, trimethylamine, and trimethylamine n-oxide (TMAO) (see *Supplementary file 2* for a full list of all metabolites obtained from the $^{1}$H NMR analysis), but we detected no significant changes in their concentration among the two pH treatments or mussel categories (*Figure 3*). Likewise, there were no significant differences in the concentration of any amino acids among mussel categories or pH treatment (*Table 1*).

Thirteen fatty acids contributed ~90% of the variation in membrane composition between the control and low pH treatment (*Table 2*). While the fatty acid profiles in intertidal *M. trossulus* were unaffected by low pH exposure (*Table 2*, *Figure 4*), the low pH treatment caused an increase in the amount of monounsaturated fatty acids (MUFA) in subtidal *M. galloprovincialis* and *M. trossulus*, and a decrease in polyunsaturated fatty acids (PUFA), in subtidal *M. trossulus* (*Figure 4*). This resulted in a significant decrease in the degree of unsaturation (i.e. lower number of double bonds in the membrane) (GLM; p<0.05; *Table 2*; *Figure 4*). Accordingly, the unsaturation index (UI; the index for the number of double bonds per 100 molecules of fatty acids) was significantly higher in intertidal than subtidal *M. trossulus* after pH exposure (Tukey's HSD, p=0.0002).

## Seawater chemistry

Mean pH measurements from the hand-held pH meter (measured on the total hydrogen ion scale; $pH_T$) showed relatively good agreement with our target acidified treatment (7.5 $pH_T$ ± 0.03–0.06 SD), although there was a greater degree of variability in the control treatments (7.9 $pH_T$ ± 0.12–0.19 SD) due to fluctuating ambient partial pressure of carbon dioxide ($pCO_2$) (*Supplementary file 3*; *Figure 5*). There was also some disagreement between our measured $pH_T$ and pH calculated using dissolved inorganic carbon (DIC) and $pCO_2$ data (*Supplementary file 4*). The discrepancies in our control treatments were the result of the highly variable ambient $pCO_2$ and the corresponding adjustments we frequently made to our gas delivery system. On average, these fluctuations did not cause significant deviations from our target pH values, as shown by our hand-held $pH_T$ data (*Supplementary file 3*) but were more pronounced in the data taken from discrete, single time point water samples. In all but one instance, the difference between directly measured and calculated pH for a single time point, that is, the direct $pH_T$ measurement made at the time the discrete water sample was taken, was within the standard deviation of mean pH measurements taken over the 2-wk period. The regulation of

**Table 1.** Mean (± SD) amino acid composition (% total amino acid content) of gill tissue in intertidal *Mytilus trossulus,* subtidal *M. trossulus,* and subtidal *M. galloprovincialis* after 10 days of acclimation to pH conditions (pH = 7.9 or pH = 7.5, n = 5).

| Amino acid | Intertidal – *M. trossulus* | | Subtidal – *M. trossulus* | | Subtidal – *M. galloprovincialis* | |
|---|---|---|---|---|---|---|
| | Acidified | Control | Acidified | Control | Acidified | Control |
| Asx | 11.18 ± 0.44 | 11.39 ± 0.25 | 11.07 ± 0.30 | 11.34 ± 0.86 | 11.07 ± 0.30 | 11.01 ± 0.33 |
| Ala | 5.50 ± 0.22 | 5.74 ± 0.35 | 5.54 ± 0.21 | 5.47 ± 0.33 | 5.54 ± 0.21 | 5.56 ± 0.13 |
| Arg | 7.92 ± 0.85 | 7.11 ± 0.10 | 7.07 ± 0.07 | 7.08 ± 0.11 | 7.07 ± 0.07 | 7.22 ± 0.19 |
| Glx | 14.08 ± 0.99 | 14.97 ± 0.16 | 15.07 ± 0.12 | 14.81 ± 0.46 | 15.07 ± 0.12 | 14.97 ± 0.17 |
| Gly | 9.37 ± 0.91 | 9.15 ± 0.62 | 9.15 ± 0.61 | 9.39 ± 1.43 | 9.15 ± 0.61 | 9.31 ± 0.7 |
| His | 2.03 ± 0.04 | 2.15 ± 0.12 | 2.04 ± 0.07 | 2.09 ± 0.10 | 2.04 ± 0.07 | 2.05 ± 0.05 |
| Ile | 4.16 ± 0.18 | 4.14 ± 0.09 | 4.03 ± 0.16 | 4.10 ± 0.26 | 4.03 ± 0.16 | 3.97 ± 0.19 |
| L-Dopa | 0.14 ± 0.01 | 0.17 ± 0.02 | 0.09 ± 0.01 | 0.14 ± 0.03 | 0.09 ± 0.01 | 0.11 ± 0.01 |
| Leu | 6.49 ± 0.18 | 6.53 ± 0.09 | 6.30 ± 0.25 | 6.42 ± 0.34 | 6.30 ± 0.25 | 6.3 ± 0.3 |
| Lys | 7.65 ± 1.60 | 7.06 ± 0.33 | 6.51 ± 0.37 | 6.69 ± 1.16 | 6.51 ± 0.37 | 6.47 ± 0.36 |
| Met | 2.55 ± 0.12 | 2.60 ± 0.08 | 2.76 ± 0.05 | 2.64 ± 0.10 | 2.76 ± 0.05 | 2.68 ± 0.08 |
| Phe | 4.26 ± 0.19 | 4.36 ± 0.12 | 4.16 ± 0.15 | 4.31 ± 0.44 | 4.16 ± 0.15 | 4.08 ± 0.18 |
| Pro | 5.25 ± 0.48 | 5.26 ± 0.27 | 5.92 ± 0.31 | 5.59 ± 0.61 | 5.92 ± 0.31 | 6.03 ± 0.24 |
| Ser | 5.06 ± 0.21 | 4.85 ± 0.15 | 4.81 ± 0.10 | 4.87 ± 0.14 | 4.81 ± 0.10 | 4.77 ± 0.1 |
| Thr | 4.36 ± 0.16 | 4.38 ± 0.04 | 4.72 ± 0.07 | 4.50 ± 0.13 | 4.72 ± 0.07 | 4.75 ± 0.05 |
| Tyr | 5.12 ± 0.47 | 5.25 ± 0.28 | 5.62 ± 0.10 | 5.55 ± 0.44 | 5.62 ± 0.10 | 5.63 ± 0.15 |
| Val | 5.03 ± 0.24 | 5.06 ± 0.11 | 5.22 ± 0.10 | 5.13 ± 0.11 | 5.22 ± 0.10 | 5.21 ± 0.08 |

Asx = aspartic acid/asparginine. Ala = alanine. Arg = arginine. Glx = glutamic acid/glutamine. Gly = glycine. His = histidine. Ile = isoleucine. L-Dopa. Levodopa; Leu = leucine. Lys = lysine. Met = methionine. Phe = phenylalanine. Pro = proline. Ser = serine. Thr = threonine. Tyr = tyrosine. Val = valine.

ocean acidification simulation systems with potentiometric pH meters has been shown to be reliable (**MacLeod et al., 2015**), and therefore it is likely that the discrepancy between discrete and hand-held $pH_T$ data was not indicative of substantial deviations in seawater chemistry target values.

The addition of $CO_2$ free air to the controls also resulted in lower-than-expected $pCO_2$ values in both start and end point data from those treatments (**Supplementary files 4 and 5**). We also observed some anomalous values for end point total alkalinity and DIC in our control treatment that were attributed to shell calcification and insufficient water replacement rates (**Supplementary files 4 and 5**). These values were not indicative of the seawater chemistry parameters over the entire experimental period, as described above, but are included for completeness.

## Seawater chemistry variability

In contrast to open oceans where pH is stable (**Hofmann et al., 2011**), daily and seasonal fluctuations can exceed 0.7 pH units in coastal ecosystems (**Baumann et al., 2015**; **Menéndez et al., 2001**; **Wootton et al., 2012**; **Wootton et al., 2008**), where dense blue mussel beds have been found in areas with a seawater saturation level of aragonite calcium carbonate ($\Omega_{arag}$) below 0.5 (**Duarte et al., 2020**). $\Omega_{arag}$ control calcification kinetics and $\Omega_{arag}$ <1 means conditions are corrosive for aragonite-based shells, like blue mussels. Along the intertidal rocky shoreline of the Northwest Pacific, where mussels for this study were collected, pH values naturally decline below 7.6, particularly during the winter (**Ianson et al., 2016**; **Jarníková et al., 2022**; **Simpson et al., 2022**). Thus, while pH conditions in our aquariums fluctuated (control 7.84 ± 0.19–7.92 ± 0.12 I acidified 7.49 ± 0.05–7.55 ± 0.03), the variability was within the range of in situ fluctuation rates. Our control conditions therefore represent actual in situ conditions, and the final average difference in pH between the control (7.88) and

**Table 2.** Mean (± SD) fatty acid composition (% fatty acid content) of the 13 fatty acids contributing ~90% of the differences in fatty acid composition after 10 days of acclimation to pH conditions (pH = 7.9 or pH = 7.5).

UI = unsaturation index. Bold numbers indicate significant differences among the control and low pH treatment within a mussel category (ANOVA, p<0.05, n = 5).

| Fatty acid | Intertidal – *M. trossulus* | | Subtidal – *M. trossulus* | | Subtidal – *M. galloprovincialis* | |
|---|---|---|---|---|---|---|
| | Acidified | Control | Acidified | Control | Acidified | Control |
| 16:0 | 9.53 ± 0.89 | 8.72 ± 1.21 | 16.56 ± 6.62 | 10.21 ± 1.21 | 12.95 ± 5.17 | 8.83 ± 1.44 |
| 18:0 | 2.57 ± 0.34 | 2.53 ± 0.17 | 4.30 ± 1.98 | 2.28 ± 0.38 | 3.38 ± 1.26 | 2.53 ± 0.67 |
| 16:1ω7 | 1.01 ± 0.1 | 1.2 ± 0.22 | **1.58 ± 0.33** | **0.97 ± 0.19** | **1.71 ± 0.33** | **1.07 ± 0.27** |
| 17:1ω7 | 1.71 ± 0.63 | 4.07 ± 4.16 | 1.94 ± 1.14 | 1.43 ± 1.1 | 2.26 ± 1.63 | 2.04 ± 2.15 |
| 18:1ω7 | 1.48 ± 0.12 | 1.37 ± 0.37 | **1.91 ± 0.52** | **1.21 ± 0.2** | 2.08 ± 0.58 | 1.5 ± 0.34 |
| 20:1ω9 | 4.31 ± 0.3 | 4.12 ± 0.46 | **6.21 ± 1.13** | **4.86 ± 0.61** | 5.43 ± 1.32 | 4.37 ± 0.48 |
| 18:2ω6 | 0.91 ± 0.1 | 0.94 ± 0.35 | 1.46 ± 0.67 | 1.31 ± 0.25 | 0.79 ± 0.36 | 0.64 ± 0.14 |
| 18:3ω3 | 1.12 ± 0.07 | 1.21 ± 0.5 | 1.04 ± 0.57 | 1.52 ± 0.19 | 0.31 ± 0.1 | 0.36 ± 0.09 |
| 20:2ω6 | 9.41 ± 0.76 | 8.82 ± 1.05 | **11.96 ± 1.21** | **9.87 ± 0.58** | 9.78 ± 1.79 | 8.67 ± 1.41 |
| 20:4ω6 | 9.70 ± 1.1 | 9.81 ± 0.9 | 3.76 ± 2.61 | 6.07 ± 0.76 | 6.95 ± 2.43 | 9.7 ± 1.37 |
| 20:5ω3 | 9.78 ± 1.15 | 10.74 ± 1.97 | **4.87 ± 3.59** | **12.19 ± 1.05** | 7.0 ± 2.98 | 10.21 ± 1.68 |
| 22:2-NMI | 11.54 ± 0.84 | 10.86 ± 1.5 | **12.72 ± 0.95** | **9.55 ± 0.66** | 13.53 ± 1.5 | 12.19 ± .1.14 |
| 22:6ω3 | 14.13 ± 0.99 | 13.73 ± 0.69 | **6.36 ± 4.86** | **16.22 ± 0.48** | **9.7 ± 4.61** | **15.88 ± 2.06** |
| UI | 234.73 ± 6.14 | 238.84 ± 8.75 | **150.94 ± 55.19** | **244.97 ± 4.59** | **187.59 ± 46.35** | **242.92 ± 8.78** |

16:0, palmitic acid; 18:0, stearic acid; 16:1ω7, palmitoleic acid; 17:1ω7, 10Z-heptadecenoic acid; 18:1ω7, vaccenic acid; 20:1ω9, gondoic acid; 18:2ω6, linolelaidic acid; 18:3ω3, γ-linolenic acid; 20:2ω6, docosadienoic acid; 20:4ω6, arachidonic acid; 20:5ω3, eicosapentaenoic acid; 22:2-NMI, non-methylene-interrupted dienoic acid; 22:6ω3, docosahexaenoic acid.

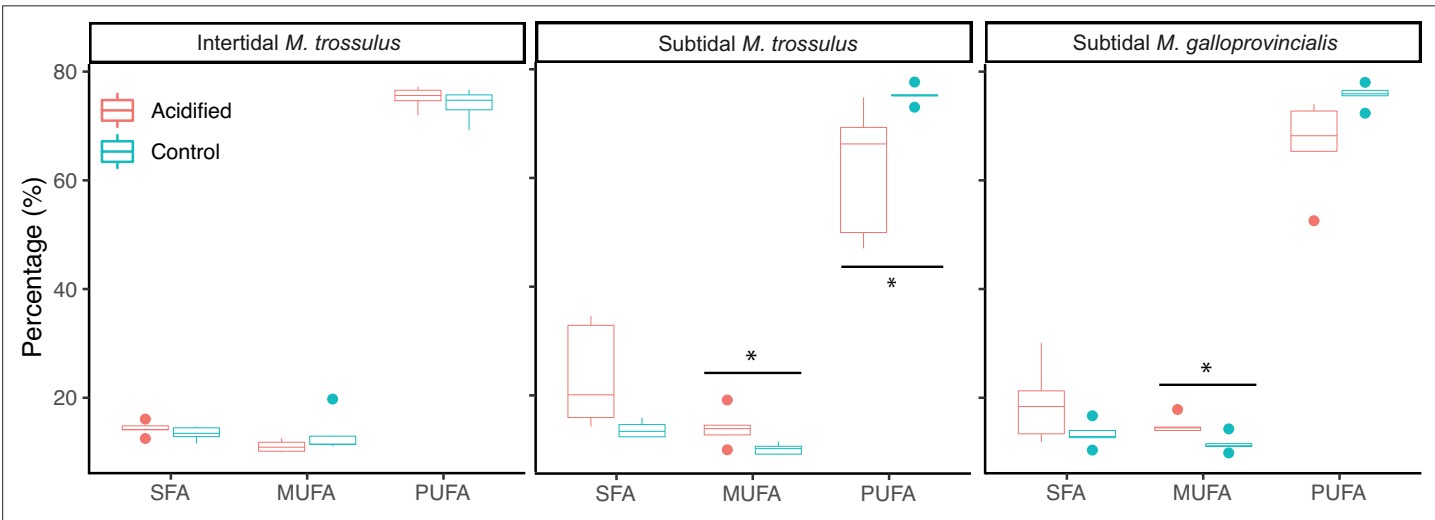

**Figure 4.** The molar percentage of saturated fatty acids (SFA), monounsaturated fatty acids (MUFA), and polyunsaturated fatty acids (PUFA) of gill tissue in intertidal *Mytilus trossulus,* subtidal *M. trossulus,* and subtidal *M. galloprovincialis* after 10 days of acclimation to pH conditions (pH = 7.9 or pH = 7.5). The horizontal line in each boxplot is the median, the boxes define the hinges (25–75% quartile), and the whisker is 1.5 times the hinges (n = 5). Colored dots represent data outside this range. Asterisks indicate a significant difference (ANOVA, p<0.05) between control and acidified conditions.

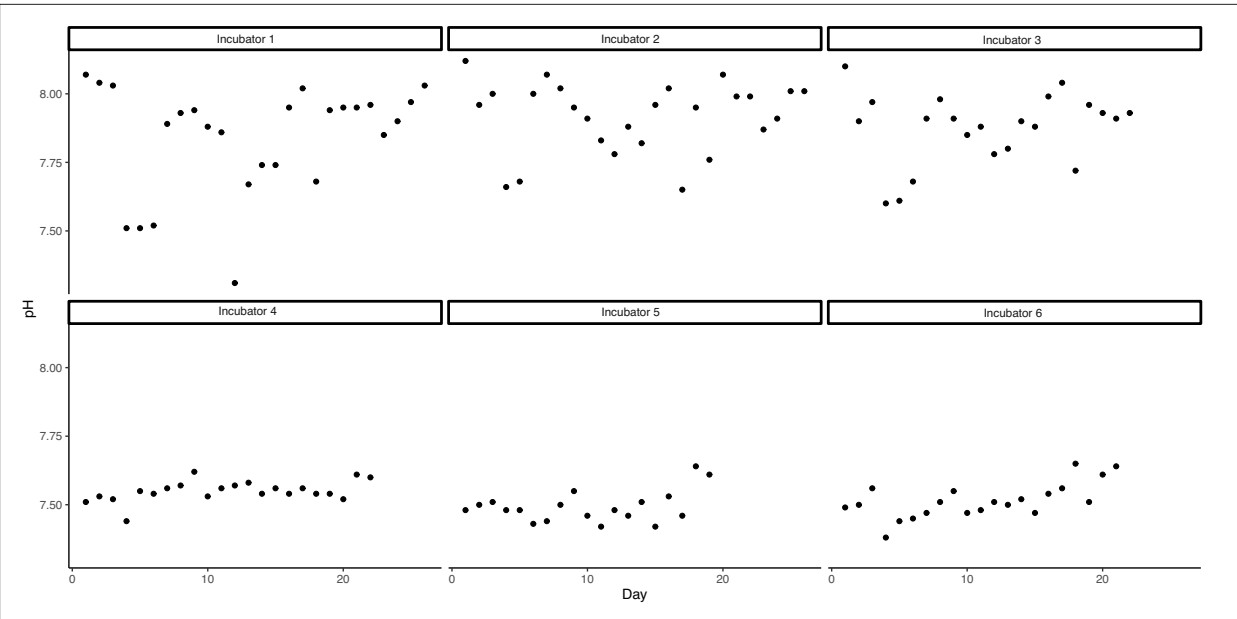

**Figure 5.** pH measurements from each incubator during the observation period. Incubators 1–3 were set to control conditions (pH = 7.9), while incubators 4–6 were set to acidified conditions (pH = 7.5).

acidified (7.52) represented our target values (7.9 pH and 7.5 pH). The natural variation in coastal pH also challenges the common belief that $\Omega_{arag}$ should be >1 (non-corrosive conditions) to represent control conditions in ocean acidification experiments. Widespread undersaturation of surface water aragonite ($\Omega_{arag}$< 1) occurs in the waters in the Strait of Georgia during winter (**Simpson et al., 2022**), and we argue that control conditions should reflect actual pH levels on the site of collections, regardless of the $\Omega_{arag}$ level. While we acknowledge that our regulation of seawater chemistry could be improved, we believe that changes in mussel survival were caused by changes in average pH, rather than variation in pH or other parameters. Our rationale is supported by the fact that intertidal mussels exhibited the largest decrease in survival upon exposure to reduced pH plus freezing, and as they are typically exposed to much greater variability in seawater chemistry and temperature than subtidal mussel populations, it is highly unlikely that variability had the most pronounced and negative effect on this category.

## Discussion

Climate change is redistributing species toward cooler environments but understanding how different drivers interact to shape species geographical ranges is essential for predicting patterns and rates of change. At higher latitudes, range-expanding species face a suite of novel abiotic conditions including low temperatures and a decreasing seawater pH (**Fassbender et al., 2017**). The goals of this study were to investigate the combined effect of low seawater pH and sub-zero air temperature stress on survival of two *Mytilus* spp. and compare the responses between a native and invasive congener. Intertidal individuals of the native bay mussel *M. trossulus* were significantly more freeze tolerant than subtidal *M. trossulus* individuals, which were in turn more freeze tolerant than the invasive Mediterranean mussel *M. galloprovincialis*. Following exposure to acidified seawater, our data demonstrated a significant negative effect on freeze tolerance and survival across all species-habitat combinations. Interestingly, the intertidal population of *M. trossulus* was most impacted by acidification, while subtidal *M. trossulus* was the least affected, and was more freeze tolerant than the other categories exposed to acidified seawater. Cellular accumulation of metabolites and reconfiguration of membrane fatty acids were uncorrelated with the observed variation in survival among mussel categories under both control and acidified conditions, which could be related to short-term exposure.

Under present-day conditions (our control pH treatment), both the intertidal and subtidal *M. trossulus* category were more freeze tolerant than the invasive *M. galloprovincialis*. This corresponds to

their geographical distribution with *M. trossulus* predominantly inhabiting shorelines at higher latitudes where winter sub-zero air temperatures are common, while *M. galloprovincialis* dominate on warmer low-latitude shores (*Hilbish et al., 2000*). However, the physiological processes behind inter- and intraspecific variation in freeze tolerance remain poorly understood. In *Mytilus* sp., the accumulation of intracellular low molecular weight osmolytes increases freeze tolerance (*Kennedy et al., 2020*; *Williams, 1970*), but the accumulation of these putative cryoprotectants can only partly explain survival after sub-zero temperature exposure. For example, although individuals of *M. trossulus* living in the upper intertidal zone are more freeze tolerant than individuals from the low zone, a recent study showed no differences in the concentration of metabolites among the shore levels (*Kennedy et al., 2020*), and no differences in the concentration of putative cryoprotectants were observed among our three mussel categories, despite large variation in freeze tolerance. Likewise, after 10 days of exposure to acidified water that significantly reduced freeze tolerance in all three mussel categories, with the survival of the intertidal population most affected, the low pH exposure had no effect on metabolite concentration in any mussel category, offering no explanation for the observed decrease in freeze tolerance. The fact that the intertidal category was most affected by low pH supports our notion that decreased survival was caused by changes in average pH, and not pH variation (see section 'Seawater chemistry variability') because animals from more unstable environments (i.e. the intertidal) are generally more resilient to changing environmental conditions (*Clark et al., 2018*).

The composition of the membrane's phospholipids is also proposed to determine freeze tolerance. Specifically, a positive relationship between survival and membrane unsaturation state has been shown in some species (*Bindesbøl et al., 2005*; *Slotsbo et al., 2016*). We hypothesized that freeze tolerance in *Mytilus* mussels would also be correlated to the unsaturation state. However, we observed no significant differences in the unsaturation index among mussel categories under control pH conditions. The unsaturation index in subtidal *M. trossulus* and *M. galloprovincialis* decreased in response to ocean acidification, yet they were the least affected in terms of freeze tolerance. Meanwhile, intertidal *M. trossulus* had the highest unsaturation index, but the lowest survival. Our results suggest that phospholipid composition is of limited importance for freeze tolerance in *Mytilus* mussels, while membrane reconfiguration seems to be important for keeping membranes functional in cold water environments (*Pernet et al., 2007*; *Thyrring et al., 2017c*), thus membrane reconfiguration may be important for species to inhabit cold subtidal environments.

While we were unable to explain the variation in freeze tolerance under present-day and acidified conditions, variation in freeze tolerance among populations and congeners may be explained by high molecular weight cryoprotectants, for example, ice binding proteins, not measured here. Indeed, the influence of antifreeze proteins on freeze tolerance in *Mytilus* ought to be explored further as their potential role seems to vary among populations (*Box et al., 2022*; *Loomis, 1995*). Furthermore, thermal tolerance variation may be explained at the gene level (*Clark et al., 2021*; *Peck et al., 2015*). A recent study highlighted that differences in the expression of heat shock genes and aquaporins plays a central role in determining freeze tolerance in northern barnacles species (*Marshall et al., 2018*), and heat shock proteins have been linked to sub-zero temperature survival in insects (*Rinehart et al., 2007*). Populations from variable environments (such as the intertidal zone or polar regions) are regularly exposed to unpredictable conditions, which can introduce a front loading of stress genes that enable individuals to better cope with unfavorable conditions (*Drake et al., 2017*; *Marshall et al., 2021*). Such front loading of genes is known from other marine species (*Clark et al., 2008*; *Drake et al., 2017*), and freeze-tolerant *Mytilus* populations may also have front-loaded genes (e.g. heat shock genes, aquaporins) that are constantly at a higher expression level, which transfer into resilience through faster production of stress mediating proteins (*Barshis et al., 2013*). Thus, because the intertidal population of *M. trossulus* is acclimatized to daily air exposure, compared to subtidal *M. trossulus* and *M. galloprovincialis*, constantly increased gene expression may provide an explanation for the difference observed in survival following sub-zero air exposure. Following low pH exposure, intertidal *M. trossulus* may be the most affected because intertidal species generally already live close to their physiological limits and have a limited capacity to adapt to new conditions. The large increase in mortality following low pH exposure could indicate that accommodating this additional environmental stressor exceeds their physiological ability to cope with external stressors. A molecular investigation could reveal the processes behind variation in freeze tolerance among populations and species, and investigations

into the underlying genetic mechanisms accounting for our observations would be interesting for future research.

Overall, *Mytilus* sp. are excellent at adapting to local environments (*Riginos and Cunningham, 2005*), making them highly stress tolerant and capable of enduring large ranges of salinities and temperatures (*Barrett et al., 2022*; *Nielsen et al., 2021*). While intertidal *M. trossulus* populations are found as far north as northern Greenland (*Mathiesen et al., 2017*), the northern distribution limit of the invasive *M. galloprovincialis* in the Northwest Pacific is set around Canada. At their range edge in the waters of British Columbia, Canada, subtidal populations face intense predation by seastars, excluding mussels from the subtidal and low intertidal (*Harley, 2011*). The low pH scenario tested here (pH = 7.5) revealed that acidification weakens freeze tolerance across *Mytilus* spp. Because winter low tides predominantly occurs at nighttime in the Northwest Pacific, occasionally exposing sessile intertidal organisms to air temperatures down to –10°C (*Kennedy et al., 2020*), significant annual freeze mortality events could occur in both species inhabiting the intertidal if pH continues to decline. Thus, an ongoing poleward expansion in the intertidal (where predation is less intense) could be hindered, offsetting the poleward expansion predicted because of warming waters. Consequently, if temperatures become too high for survival at a species equatorward edge, the combined effects of predation and limited freeze tolerance could result in a range contraction (rather than a latitudinal shift), substantially threatening the persistence of these species in some regions.

## Materials and methods
### Animal collection and holding conditions
Three categories of *Mytilus* mussels were collected on December 8–10, 2019, in the strait of Georgia, British Columbia, Canada; (1) subtidal *M. galloprovincialis* obtained from an aquaculture farm at Saltspring Island (48.731–123.429), (2) subtidal *M. trossulus* collected from floating docks at the Jericho Royal Vancouver Yacht Club (49.276–123.186) in the Burrard Inlet, and (3) intertidal *M. trossulus* collected at low tide from Tower Beach (49.273–123.258) in the Burrard Inlet (collection permit number XMCFR 7 2019; Fisheries and Oceans Canada). Intertidal *M. galloprovincialis* was not considered as no intertidal populations are established in the region. All mussels were kept for a 72 hr adjustment period in aerated aquaria of similar environmental conditions as the individual collection sites measured on the days of collection (7°C, pH = 7.9, and salinity 20.5). No mussels died during the adjustment period.

Two pH conditions were selected to cover a realistic range of pH values currently observed or predicted for the Southern Strait of Georgia; a control (pH = 7.9) and a low pH treatment (pH = 7.5) (*Ianson et al., 2016*). Prior to sub-zero air temperature exposure, mussels were maintained in low or control pH conditions for 10 days using three incubators (Panasonic MIR 154, Panasonic, Japan) for low pH conditions and three for control conditions (no mussels died during the 10 days). Each incubator was set to 7°C and contained three 5 L glass aquaria that held three envelopes of 0.5 cm gauge, rigid plastic mesh (25 × 24 cm) that separated the three categories of mussel but allowed unrestricted circulation of seawater (salinity 20–21).

### Seawater manipulation
The low pH treatment was established using two Smart-Trak mass flow controllers (Sierra Instruments, Inc, CA) to mix 100% $CO_2$ (PraxAir Canada Inc, CB, Canada) and $CO_2$-free air, which was then bubbled into the acidified seawater aquaria to achieve target values of 7.50 pH. $CO_2$-free air was generated using a small compressor to pump ambient air through a 500 mL Nalgene canister that contained Soda Lime (Ormond Veterinary Supply Ltd., ON, Canada). A flow rate of 3.3 cm$^3$/s of 100% $CO_2$ gas and 4.11 L/min of $CO_2$-free air was used to reach the target pH. Our system also removed moisture from ambient air to protect the mass flow controllers from water damage. This was achieved by running the ambient-air gas lines through a small refrigerator to reduce air temperature and cause water to precipitate into a water trap, and by installing a second 500 mL Nalgene containing desiccant (WA Hammond Drierite, OH) in series with the soda lime container. Control conditions (7.90 pH) were maintained by mixing ambient air and $CO_2$-free air. The use of $CO_2$-free air was necessary as ambient air was artificially high in $CO_2$ due to poor ventilation in the lab. As with the low pH treatment, ambient air was pumped through a Soda Lime-filled 500 mL Nalgene canister using a small

compressor before being connected to a three-way splitter and bubbled into seawater aquaria, while ambient air was bubbled into the control aquaria using a second set of tubes connected to low-power aquarium air pumps (Fusion 700 Air Pump). Air flow from the small aquarium pumps was fine-tuned by placing an adjustable clamp on the flexible tubing that connected pump and air stone to increase or decrease the flow of $CO_2$-enriched ambient air to achieve 7.90 pH. Carbon dioxide in ambient 'lab' air was monitored constantly using a Qubit S151 CO2 gas analyzer (Qubit Systems, ON, Canada), which showed that $CO_2$ fluctuated during the day between 400 ppm and 600 ppm $CO_2$ reaching the maximum during the day while people were working in the lab. Consequently, the input of ambient air into control tanks was monitored and adjusted daily (mainly during the day) to maintain target pH values. Prior to adjusting seawater pH, we mixed filtered seawater (provided by the Vancouver Aquarium and transported by the City of Vancouver) with de-chlorinated distilled fresh water to create a 20–21 ppt solution that was the salinity recorded at the collection sites. Mussels were able to feed on phytoplankton naturally occurring in the water, and we replaced 50% of the seawater from each tank daily to prevent the buildup of feces and maintain uniform seawater chemistry parameters.

## Carbonate chemistry

Seawater pH was measured daily in all aquaria using a hand-held pH meter (*Supplementary file 3*; Oakton pH 450 (± 0.01 pH), Oakton Instruments, IL) calibrated with two saltwater buffers, as described in *MacLeod et al., 2015* to provide pH measurements on the total hydrogen ion scale ($pH_T$). To further characterize the seawater carbonate chemistry, seawater samples (300 mL) were collected from one randomly selected aquarium in each incubator at the start and end of the experiment. These samples were fixed with a saturated solution of mercuric chloride (RICCA Chemical Company, TX) and analyzed using the 'burke-o-lator' at the Hakai Institute (Quadra Island, BC, Canada); for details of this system, see *Evans et al., 2019*. This analysis generated values for DIC and $pCO_2$, which were then used in combination with temperature and salinity data to calculate all relevant carbonate parameters (*Supplementary file 5*) using the MATLAB version of CO2SYS (*van Heuven et al., 2011*).

## Sub-zero temperature exposure

After 10 days of acclimation to pH conditions (pH = 7.9 or pH = 7.5), groups of mussels were exposed to seven sub-zero air temperatures (−5, –6, −7,–8, −9,–12, and –15°C) for 2 hr by placing animals in individual plastic tubes inserted in wells drilled into a precooled aluminum block cooled by refrigerated circulation baths (Thermo Fisher Scientific Inc, MA; *Figure 1—figure supplement 1*). Fifteen mussels (mean shell length 37.69 mm ± 3.14 SD) from each mussel category (subtidal *M. galloprovincialis* and *M. trossulus*, and intertidal *M. trossulus*) and pH condition (pH = 7.9 and pH = 7.5) were used at every temperature for a total of 720 mussels. Individual body temperatures were recorded at 0.5 s intervals using Type-T thermocouples (Omega, QC, Canada) placed next to the shell inside the plastic tube and connected to TC-08 thermocouples interfaces (Pico Technology, UK) that interfaced to a computer running PICOLOG 6 beta software (Picotech, UK), which continuously monitored body temperatures. Continuous body temperature monitoring allowed us to determine any exothermic release of heat owing to ice formation. The lowest temperature prior to this event is termed the supercooling point (SCP), and the SCP indicates that internal ice formation occurred (*Sinclair et al., 2015*). After 2 hr of sub-zero air exposure, all mussels were transferred back to their respective prefreezing pH condition aquaria for recovery where they were monitored daily for 5 days to record mortality. Mortality was checked daily with mussels considered dead if they did not close their shells when touched. Dead mussels were immediately removed from the aquaria and had their shell length measured to nearest mm.

## Metabolite analysis

All metabolite and fatty acid analyses (next section) were performed on gill tissue because this tissue is essential for gas exchange, metabolism, and is directly exposed to internal ice formation (*Kennedy et al., 2020*; *Pernet et al., 2007*). Total amino acid analysis was performed on mussels collected after 10 days of pH exposure (mean dry weight = 15.46 mg ± 0.61 SD, n = 5) at the SickKids Proteomics, Analytics, Robotics & Chemical Biology Centre (SPARC; https://lab.research.sickkids.ca/sparc-molecular-analysis/services/amino-acid-analysis/), Hospital for Sick Children, Toronto, ON, Canada, using the Water Pico-Tag System (Water Corporation, WA). The final concentration of each amino acid was

calculated in µg·mg⁻¹ and then expressed as relative concentration (% of total amino acids). It should be noted that this amino acid analysis did not allow discrimination between Asn/Asp and Gln/Glu.

One-dimensional, 600 MHz proton nuclear magnetic resonance spectroscopy ($^1$H NMR) was used to measure profiles of other soluble molecules in the gill tissue (collected after 10 days of pH exposure). $^1$H NMR is ideal for measuring low molecular weight, polar metabolites such as osmolytes and anaerobic byproducts. Sample preparation was based on *Cappello et al., 2013*. A 100 mg sample of gill tissue was excised (n = 5), dried with a Kimwipe to remove excess water and frozen at −80°C. Frozen tissue was homogenized in 400 µL cold methanol and 85 µL cold water-xylitol solution (5 mM xylitol as an internal control) using a bead homogenizer (Bullet Blender 50 Gold Model: BBX24, Next Advance) with approximately 200 µL of 3.2 mm round stainless steel beads, for 10 min at setting 8 in 1.5 mL microcentrifuge vials. After adding 400 µL chloroform and 200 µL water to the samples, they were vortexed for 60 s, left on ice for 10 min for phase separation, and centrifuged for 5 min at 2000 rpm. The upper methanol layer (600 µL) containing the polar metabolites was transferred into new vials, dried in a centrifugal vacuum concentrator (Eppendorf 5301), and then stored at −80°C. Immediately prior to $^1$H NMR analysis, the dried polar extracts were resuspended in 600 µL of 0.1 mol/L sodium phosphate buffer (pH 7.0, 50% deuterium oxide, Sigma-Aldrich) containing 1 mmol/L 2,2-dimethyl-2-sila-pentane-5-sulfonate (DSS; Sigma-Aldrich) as internal reference. The mixture was vortexed for 60 s and transferred to a 5 mm NMR tube.

$^1$H NMR spectra were acquired using Bruker Avance 600 with cryoprobe and Bruker Avance III 600 spectrometers. TopSpin software version 2.1 (Bruker) was used to process spectra collected with the Bruker Avance 600 spectrometer with cryoprobe, and TopSpin version 3.5 (Bruker) was used with the Bruker Avance III 600 spectrometer. Experiments required 15 min of acquisition time and were performed at room temperature.

Peak identification of the NMR spectra was performed with Chenomx NMR Suite 9.0 (Chenomx, AB, Canada) that uses the Human Metabolome Database compound spectral reference library. First, line broadening of 2.5 Hz, automatic phase correction, and manual baseline correction were performed with Chenomx Processor (within the Chenomx NMR Suite software). Then, determination of metabolite concentrations was performed using Chenomx Profiler, which determines the concentrations of individual metabolites using the concentration of a known DSS signal. Metabolite concentrations are reported as mmol/100 mg gill wet mass.

## Fatty acid analysis

Fatty acid analysis was also conducted on mussels collected after 10 days of pH exposure (mean wet weight = 0.38 g ± 0.1 SD, n = 5). Total lipids were extracted by grinding in a dichloromethane:methanol (2:1, v/v) solution following a slightly modified Folch procedure (*Parrish, 1999*). Lipid extracts were separated into neutral and polar fractions by column chromatography on silica gel micro-columns (30 × 5 mm i.d., packed with Kieselgel 60, 70–230 mesh; Merck, Germany) using chloroform:methanol (98:2, v/v) to elute neutral lipids, followed by methanol to elute polar lipids (*Marty et al., 1992*). Fatty acid profiles were determined on fatty acid methyl esters (FAMEs) using sulfuric acid:methanol (2:98, v/v) and toluene. FAMEs of neutral and polar fractions were concentrated in hexane, and the neutral fraction was purified on an activated silica gel with 1 mL of hexane:ethyl acetate (1:1 v/v) to eliminate free sterols. FAMEs were analyzed in the full-scan mode (ionic range: 60–650 m/z) on a Polaris Q ion trap coupled multi-channel gas chromatograph 'Trace GC ultra' (Thermo Scientific, USA) equipped with an autosampler model Triplus, a PTV injector, and a mass detector model ITQ900 (Thermo Scientific). The separation was performed with an Omegawax 250 (Supelco) capillary column with high-purity helium as a carrier gas. Data were treated using Xcalibur v.2.1 software (Thermo Scientific). Methyl nondecanoate (19:0) was used as an internal standard. FAMEs were identified and quantified using known standards (Supelco 37 Component FAME Mix and menhaden oil; Supleco) and were further confirmed by mass spectrometry (Xcalibur v.2.1 software).

## Statistical analysis
### Survival

Statistical analysis was performed using the R software (R version 3.5.2). A logistic regression model was used to calculate LLT$_{50}$ values (the lower lethal temperature where 50% of the population survived). A binomial GLM with a logit link function was used to determine the effects of air temperature and

pH treatment on survival within each mussel category, and the differences in $LLT_{50}$ were estimated using 95% confidence intervals (CI) with non-overlapping CI indicating a significant difference (*Deere et al., 2006*). Differences in the SCP among mussel categories and pH treatment was analyzed using a two-way ANOVA. Final models were validated by plotting residuals versus fitted values, versus each covariate in the model (*Zuur et al., 2016*). Validation of ANOVAs and GLM models indicated no violation of model assumptions.

## Metabolomics and fatty acids

Generalized linear models and ANOVAs were used to determine which metabolites and fatty acids differed significantly after low pH exposure. The fatty acids explaining most of the dissimilarity between mussel categories and pH treatments were identified using a SIMPER analysis (see the full list of fatty acids in *Supplementary file 6*). This analysis revealed that 13 fatty acids explained ~90% of the Bray–Curtis dissimilarity amongst fatty acid profiles between the control and low pH environment (*Table 2*). We therefore focused all subsequent fatty acid analyses on these 13 fatty acids. PCA was used to interpret differences in the metabolomic composition among mussel categories. ANOVAs were used to evaluate differences in the concentrations of amino acids, and GLMs to evaluate the distribution of saturated fatty acids (SFA), monounsaturated fatty acids (MUFA), polyunsaturated fatty acids (PUFA), and the unsaturation index (UI), among the three mussel categories and pH treatment. Post hoc pair-wise tests were used to compare significant treatment effects ($p < 0.05$). Detailed data exploration was carried out prior to any analysis (*Zuur et al., 2010*). Once valid models were identified, we re-examined the residuals to ensure all model assumptions were acceptable.

## Acknowledgements

We acknowledge the technical assistance and expertise of researchers at the Hakai Institute, Quadra Island, British Columbia, who conducted the chemical analysis of our discrete seawater samples, and Vancouver Aquarium and by the City of Vancouver for providing water and transportation. This research was supported by a Marie Sklodowska-Curie Individual Fellowship (IF) under contract number 797387, the Independent Research Fund Denmark (Danmarks Frie Forskningsfond) (DFF-International Postdoc; case no. 7027-00060B), the Carlsberg Foundation (case no. CF21-0564) to JT, a Canada Foundation for Innovation Leaders Opportunity Fund grant to CDGH, and individual NSERC Discovery grants to KEM (case no. RGPIN-2019-04239) and CDGH (case no. RGPIN-2016-05441). The funding sources were not involved in designing the study design, in the collection, analysis and interpretation of data, in the writing of the report, or in the decision to submit the article for publication. We acknowledge that the University of British Columbia Vancouver campus is situated on the traditional, ancestral, and unceded territory of the Musqueam people.

## Additional information

### Funding

| Funder | Grant reference number | Author |
| --- | --- | --- |
| H2020 Marie Skłodowska-Curie Actions | 797387 | Jakob Thyrring |
| Danmarks Frie Forskningsfond | 7027-00060B | Jakob Thyrring |
| Natural Sciences and Engineering Research Council of Canada | RGPIN-2019-04239 | Katie E Marshall |
| Carlsbergfondet | CF21-0564 | Jakob Thyrring |
| Natural Sciences and Engineering Research Council of Canada | RGPIN-2016-05441 | Christopher DG Harley |

| Funder | Grant reference number | Author |
| --- | --- | --- |
| Canada Foundation for Innovation | Leaders Opportunity | Christopher DG Harley |

The funders had no role in study design, data collection and interpretation, or the decision to submit the work for publication.

## Author contributions

Jakob Thyrring, Conceptualization, Data curation, Formal analysis, Funding acquisition, Validation, Investigation, Visualization, Methodology, Writing – original draft, Project administration; Colin D Macleod, Conceptualization, Formal analysis, Validation, Investigation, Visualization, Methodology, Writing – review and editing; Katie E Marshall, Conceptualization, Supervision, Validation, Investigation, Methodology, Writing – review and editing; Jessica Kennedy, Formal analysis, Validation, Investigation, Methodology, Writing – review and editing; Réjean Tremblay, Validation, Investigation, Methodology, Writing – review and editing; Christopher DG Harley, Conceptualization, Supervision, Investigation, Methodology, Writing – review and editing

## Author ORCIDs

Jakob Thyrring ⓘ http://orcid.org/0000-0002-1029-3105
Katie E Marshall ⓘ http://orcid.org/0000-0002-6991-4957
Christopher DG Harley ⓘ http://orcid.org/0000-0003-4099-943X

## Decision letter and Author response

Decision letter https://doi.org/10.7554/eLife.81080.sa1
Author response https://doi.org/10.7554/eLife.81080.sa2

## Additional files

### Supplementary files

• Supplementary file 1. Estimated regression parameters, standard errors, z-values, and p-values for the binomial generalized linear models (GLM).

• Supplementary file 2. Mean (± SD) metabolite concentration (nmol/100 g ww gill tissue) detected from the $^1$H NMR analysis after 10 days of acclimation to pH conditions (pH = 7.9 or pH = 7.5) (n = 5).

• Supplementary file 3. Mean (± SD) total hydrogen ion scale ($pH_T$) and temperature measured directly in aquaria in each incubator.

• Supplementary file 4. Mean (± SD) carbonate measurements taken from discrete water samples at the start and end of the exposure period.

• Supplementary file 5. Carbonate measurement data from the burke-o-lator system at the Hakai Institute. Bold numbers indicate anomalous values.

• Supplementary file 6. Mean (± SD) fatty acid composition (% fatty acid content) of all fatty acids after 10 days of acclimation to pH conditions (pH = 7.9 or pH = 7.5) (n = 5). The molar percentage of saturated fatty acids (SFA), monounsaturated fatty acids (MUFA), and polyunsaturated fatty acids (PUFA) is presented.

• MDAR checklist

### Data availability

All data produced and needed to replicate the work is deposited freely on the Zenodo data repository (https://zenodo.org): https://doi.org/10.5281/zenodo.4454508.

The following dataset was generated:

| Author(s) | Year | Dataset title | Dataset URL | Database and Identifier |
|---|---|---|---|---|
| Thyrring J, Macleod CD, Marshall KE, Kennedy J, Tremblay R, Harley CDG | 2023 | Replication data for: "Ocean acidification increases susceptibility to sub-zero air temperatures in ecosystem engineers and limit poleward range shifts" | https://doi.org/10.5281/zenodo.4454508 | Zenodo, 10.5281/zenodo.4454508 |

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
