## [Editor Report]

Thyrring et al. provide convincing experimental results on the role of ocean acidification on the survival of two bivalve species. This novel work is fundamental in setting a more mechanistic understanding of the impacts of climate change on ocean species' poleward re-distribution across the globe. The major strength of this work is their usage of state-of-the-art techniques (such as metabolomics, fatty acid and amino acid analysis) to link physiological level processes to global climate change.

---

## [Decision Letter]

**Decision letter after peer review:**

Thank you for submitting your article "Ocean acidification increases susceptibility to sub-zero air temperatures in ecosystem engineers (*Mytilus* sp.): a limit to poleward range shifts" for consideration by *eLife*. Your article has been reviewed by 3 peer reviewers, one of whom is a member of our Board of Reviewing Editors, and the evaluation has been overseen by Meredith Schuman as the Senior Editor. The reviewers have opted to remain anonymous.

Essential revisions:

The reviewers nicely provide complementary advice, but they all seem to agree that the metabolite framework (and specifically, the metabolite-metabolite correlations) are of little benefit to the manuscript. So, please reinforce the theory behind this part of your manuscript, explain the logic behind the analyses, and discuss the results in a meaningful way.

*Reviewer #3 (Recommendations for the authors):*

The authors assess response to ocean acidification with three populations of mussels encompassing two species: *Mytilus trossulus* from the intertidal and subtidal and *M. galloprovincialis* from a subtidal aquaculture farm. All three species received an ambient of low pH treatment prior to a freezing treatment. The authors find species differences in freeze tolerance in mussels, with intertidal *M. galloprovincialis* showing the least freeze tolerance. The authors assess the metabolic capacity and molecular components with the analysis of amino acids, fatty acids, and osmolytes and anaerobic byproducts.

The idea that species in fluctuating environments (here, the intertidal) might respond differently to those in constant environments (here, the subtidal) has been explored in multiple systems. I thought the authors should expand from mostly self-citations (e.g. Line 452) and include other studies. In mussels, for example, (Bitter et al. 2021, doi: 10.1098/rspb.2021.0727 & doi: 10.1086/712930). Or more general refs, e.g. Scheiner 1993. doi: 10.1146/annurve.es.24.110193.000343.

The authors hypothesized metabolic changes due to OA and cold temperatures, yet they demonstrated a significant amount of stasis with high similarity among species at the molecular level. The fatty acids in the intertidal M trossulus, the most freeze tolerant, did not change. Further, there is little explanation of molecular/metabolic changes that could explain their results. Because of this somewhat unexpected lack of signal of these stressors, I would like to see an enhanced explanation of animal homeostasis. Are these plastic responses at the molecular level? What should the first responses be for an animal under stress? The authors mention previous results relating to heat stress, and I thought it would be beneficial to discuss how the lack of a molecular response to freezing is related to the strong responses seen in heat stress.

Methods. I wondered about the ambient temperatures that the authors used. I see that their setup was 20-21 ppt and their water temp was 6-7C. What were the conditions where the mussels originated, including the aquaculture set up?

L212- Could the authors provide a picture of this as supplemental material?

L344 – What determined the compounds that were scrutinized?

L391, 392 – The authors should cite empirical studies of pH change that are in proximity to where they studied (e.g. Wootton et al. 2008 doi: 10.1073/pnas.0810079105, Wootton & Pfister 2012, doi: 10.1371/journal.pone.0053396) rather than tropical studies, for example.

L400-401 – reference for this statement?

L527 – in addition to the term 'niche squeeze', the authors should use the more familiar 'range contraction'.

Table 3 – Is there a comparison to animals in their indigenous areas? Do the amino acid concentrations reflect seawater?

Figure 1 The points at approximately -8 have a large influence on these results and shift curve to imply that intertidal M trossulus survive much better.

Table S4. Are there any differences in these compounds?

[Editors' note: further revisions were suggested prior to acceptance, as described below.]

Thank you for resubmitting your work entitled "Ocean acidification increases susceptibility to sub-zero air temperatures in ecosystem engineers and limit poleward range shifts" for further consideration by *eLife*. Your revised article has been evaluated by Meredith Schuman (Senior Editor) and a Reviewing Editor.

The manuscript has been improved but there are some remaining issues that need to be addressed, as outlined below. Please revise the manuscript fully and meticulously as a third round of revisions is unlikely.

*Reviewer #2 (Recommendations for the authors):*

The authors have done a thorough job with the reviewer's recommendations, and the manuscript is in good shape. Some nitpicky comments below:

Abstract

I see that the metabolite correlation analysis information has been removed, but now metabolites are not mentioned in the abstract at all. Should there be at least one sentence about this?

Introduction

L116 – 136: This paragraph tackles two ideas – one of the mechanisms associated with freeze tolerance (homeoviscous adaptation) and the potential impact of ocean acidification on freeze tolerance mechanisms. I suggest splitting it into two paragraphs, so that each paragraph is more focused. The line "Despite this progress on the mechanisms…" would be a good topic sentence for the second paragraph.

Results

Because the results have been moved before the methods, I have two broad suggestions.

1) Start each section/paragraph of the results with a quick (one-sentence) summary of the methods that were used to obtain the results, or why that method was being used. (e.g. Your reader will not know why you measured SCPs.)

2) There are now a few abbreviations in the results that need to be defined upon first use: SFA, PUFA, DIC, pCO2, and pHT. I think you need to keep the definition of UI on L206 because otherwise the reader first sees this abbreviation in the discussion. (Alternatively, spell the term out in full in the discussion.)

Section 2.1. The first paragraph is only a single sentence. Either combine it with the next paragraph or expand it a little.

Figure 2: What does the size of each point represent? In the figure caption, suggest explaining what each point represents (data from one individual mussel?) and what the size of the point indicates. Also, are the circles representative of a particular metric/value, e.g. 95% confidence limit of each group?

Figure 3 – y-axis title should be descriptive (e.g. Concentration) not just include units

Figure 5: suggest headings are more descriptive, e.g. "Incubator 1" instead of "1." These figures don't have to be as tall as they are either – each panel looks to be twice as tall as the panels for many of your other figures.

Discussion

L308 – 312: Long sentence. Can you split it up to make the logic easier to follow?

L315 – 16: Confused about why you include this cyprinid fish example – are you trying to make a point about how long it takes to restructure membrane composition? If so, that is not totally clear.

Supplementary Files: Can you include the table/figure captions in the supplementary files themselves? It just makes life a little easier for the reader. (But no worries if *eLife* does not permit this.)

Table S5: Can you please include a footnote that defines abbreviations in the table?

*Reviewer #3 (Recommendations for the authors):*

I have read the revised manuscript and the two other reviews of Thyrring et al. and the authors' responses to all reviewers. The revised manuscript was a pleasure to read. It is clear and the response to reviewers was generally appropriate. I still, however, think that discussion points that link their results to expectations for species responses in a fluctuating (here, intertidal) versus a constant (here, subtidal) one would enhance the generality of the results, though I note the authors reduced that discussion theme based on other feedback.

---

## [Author Response]

Reviewer #3 (Recommendations for the authors):The authors assess response to ocean acidification with three populations of mussels encompassing two species: *Mytilus trossulus* from the intertidal and subtidal and *M. galloprovincialis* from a subtidal aquaculture farm. All three species received an ambient of low pH treatment prior to a freezing treatment. The authors find species differences in freeze tolerance in mussels, with intertidal *M. galloprovincialis* showing the least freeze tolerance. The authors assess the metabolic capacity and molecular components with the analysis of amino acids, fatty acids, and osmolytes and anaerobic byproducts.The idea that species in fluctuating environments (here, the intertidal) might respond differently to those in constant environments (here, the subtidal) has been explored in multiple systems. I thought the authors should expand from mostly self-citations (e.g. Line 452) and include other studies. In mussels, for example, (Bitter et al. 2021, doi: 10.1098/rspb.2021.0727 & doi: 10.1086/712930). Or more general refs, e.g. Scheiner 1993. doi: 10.1146/annurve.es.24.110193.000343.

We have significantly reduced this paragraph. Thus, we do no longer focus on fluctuating environments.

The authors hypothesized metabolic changes due to OA and cold temperatures, yet they demonstrated a significant amount of stasis with high similarity among species at the molecular level. The fatty acids in the intertidal M trossulus, the most freeze tolerant, did not change. Further, there is little explanation of molecular/metabolic changes that could explain their results. Because of this somewhat unexpected lack of signal of these stressors, I would like to see an enhanced explanation of animal homeostasis. Are these plastic responses at the molecular level? What should the first responses be for an animal under stress? The authors mention previous results relating to heat stress, and I thought it would be beneficial to discuss how the lack of a molecular response to freezing is related to the strong responses seen in heat stress.Methods. I wondered about the ambient temperatures that the authors used. I see that their setup was 20-21 ppt and their water temp was 6-7C. What were the conditions where the mussels originated, including the aquaculture set up?

Yes, the salinity 20-21 and 6-7 C represent the conditions at the collection time. This information is present in the method section:

“All mussels were kept for a 72-hour adjustment period in aerated aquaria of similar environmental conditions as the individual collection sites measured on the days of collection (7°C, pH = 7.9, and salinity 20.5).” (Line 383-386).

L212- Could the authors provide a picture of this as supplemental material?

We have added a picture of the set-up in the supplementary material. See Figure 1-supplement figure 1.

L344 – What determined the compounds that were scrutinized?

We selected the most abundant osmolytes found in the ^1^H NMR analysis for further investigation.

L391, 392 – The authors should cite empirical studies of pH change that are in proximity to where they studied (e.g. Wootton et al. 2008 doi: 10.1073/pnas.0810079105, Wootton & Pfister 2012, doi: 10.1371/journal.pone.0053396) rather than tropical studies, for example.

Thank you for the recommendation. We wanted to highlight that coastal pH can vary in all coastal areas, but we agree some regional work should be acknowledged. We have added more suitable references. (Line 230-231).

L527 – in addition to the term 'niche squeeze', the authors should use the more familiar 'range contraction'.

We now use the term range contraction instead of niche squeeze.

Table 3 – Is there a comparison to animals in their indigenous areas? Do the amino acid concentrations reflect seawater?

To our knowledge is there no reference data to *M. galloprovincialis* from their indigenous areas in the Mediterranean Sea because low temperature studies are extremely rare from this warm region. Most research from the Mediterranean Sea focuses on heat stress and the impacts of warming.

Figure 1 The points at approximately -8 have a large influence on these results and shift curve to imply that intertidal M trossulus survive much better.

Yes, there is a shift around -8C which is very interesting, and intertidal *M. trossulus* had indeed a significant higher survival.

Table S4. Are there any differences in these compounds?

We did not test significant differences in all identified compounds. Only the dominant ones included in the main work.

[Editors' note: further revisions were suggested prior to acceptance, as described below.]

Reviewer #2 (Recommendations for the authors):The authors have done a thorough job with the reviewer's recommendations, and the manuscript is in good shape. Some nitpicky comments below:AbstractI see that the metabolite correlation analysis information has been removed, but now metabolites are not mentioned in the abstract at all. Should there be at least one sentence about this?

Yes, we have added a sentence in the abstract: “Differences in the concentration of various metabolites (cryoprotectants), or in the composition of amino acids and cell membrane phospholipid fatty acids could not explain the decrease in survival” (Line 4849).

IntroductionL116 – 136: This paragraph tackles two ideas – one of the mechanisms associated with freeze tolerance (homeoviscous adaptation) and the potential impact of ocean acidification on freeze tolerance mechanisms. I suggest splitting it into two paragraphs, so that each paragraph is more focused. The line "Despite this progress on the mechanisms…" would be a good topic sentence for the second paragraph.

We have split the paragraph (Line 112). We agree this makes both paragraphs more focused.

ResultsBecause the results have been moved before the methods, I have two broad suggestions.1) Start each section/paragraph of the results with a quick (one-sentence) summary of the methods that were used to obtain the results, or why that method was being used. (e.g. Your reader will not know why you measured SCPs.)

We agree with the reviewer, and the following method summaries have been added:

Paragraph 2.1: “Following acclimation to two pH condition (pH = 7.9 or pH = 7.5), mussels were exposed to seven sub-zero air temperatures (-5, -6, -7, -8, -9, -12, -15 °C), and the supercooling point (SCP: indication of internal ice formation) was determined.” (Line 145-147).

Paragraph 2.2: “Metabolite, amino acid and fatty acid analyses were performed on mussels collected after 10 days of pH exposure.” (Line 168-169).

2) There are now a few abbreviations in the results that need to be defined upon first use: SFA, PUFA, DIC, pCO2, and pHT. I think you need to keep the definition of UI on L206 because otherwise the reader first sees this abbreviation in the discussion. (Alternatively, spell the term out in full in the discussion.)

We thank the reviewer for pointing this out. We have spelled out the full term upon first use for all abbreviated words. Please see the revised result section.

Section 2.1. The first paragraph is only a single sentence. Either combine it with the next paragraph or expand it a little.

We have expanded section 2.1:

“2.1 Survival

Following acclimation to two pH condition (pH = 7.9 or pH = 7.5), mussels were exposed to seven sub-zero air temperatures (-5, -6, -7, -8, -9, -12, -15 °C), and the supercooling point (SCP: indication of internal ice formation) was determined. There was no significant effect of pH conditions (ANOVA; p > 0.05) or mussel categories (subtidal *Mytilus trossulus* and *M. galloprovincialis*, and intertidal *M. trossulus*; ANOVA; p > 0.05) on the SCP, which indicates that internal ice formation occurred, and freezing of the tissue was observed in all mussels exposed to temperatures below -5°C. “(Line 145-151).

Figure 2: What does the size of each point represent? In the figure caption, suggest explaining what each point represents (data from one individual mussel?) and what the size of the point indicates. Also, are the circles representative of a particular metric/value, e.g. 95% confidence limit of each group?

The larger points were just the center point of the ellipse, but it had no other meaning. The sizes have not been standardized, and we have added an explanation of what the points represent. We also indicated that the ellipse is 95% confidence limits: “Figure 2: PCA plot based on all identified metabolites in intertidal *Mytilus trossulus*, subtidal *M. trossulus* and subtidal *M. galloprovincialis* after subjection to control (pH = 7.9) and low (pH = 7.5) pH treatment (n = 5). Each point represent an individual, and the ellipses extend to the 95% confidence interval or the mussel category.” (Line 834-837)

Figure 3 – y-axis title should be descriptive (e.g. Concentration) not just include units

We have updated the y-axis title: “Concentration (mmol/100g)”.

Figure 5: suggest headings are more descriptive, e.g. "Incubator 1" instead of "1." These figures don't have to be as tall as they are either – each panel looks to be twice as tall as the panels for many of your other figures.

We have updated the description to “Incubator 1” etc. and we have made the panels smaller, so they look like the other figures. Please see revised Figure 5.

DiscussionL308 – 312: Long sentence. Can you split it up to make the logic easier to follow?

We have rephrased the sentences:

“The composition of the membrane’s phospholipids is also proposed to determine freeze tolerance. Specifically, a positive relationship between survival and membrane unsaturation state (i.e., higher number of double bonds in the membrane) has been shown in some species (Bindesbøl et al., 2005; Slotsbo et al., 2016). We hypothesized that freeze tolerance in *Mytilus* mussels would also be correlated to unsaturation state. However, we observed no significant differences in unsaturation index among mussel categories under control pH conditions. The unsaturation index in subtidal *M. trossulus* and *M. galloprovincialis* decreased in response to ocean acidification, yet they were the least affected in terms of freeze tolerance.” (Line 273-281)

L315 – 16: Confused about why you include this cyprinid fish example – are you trying to make a point about how long it takes to restructure membrane composition? If so, that is not totally clear.

We agree this sentence is confusing and not relevant for this discussion. The sentence has been deleted.

Supplementary Files: Can you include the table/figure captions in the supplementary files themselves? It just makes life a little easier for the reader. (But no worries if eLife does not permit this.)

We have included the captions for supplementary files in the main document under section “Supplement Files” according to *eLife* guidelines.

Table S5: Can you please include a footnote that defines abbreviations in the table?

Done.